# A Short Series of Case Reports of COVID-19 in Immunocompromised Patients

**DOI:** 10.3390/v14050934

**Published:** 2022-04-29

**Authors:** Mitali Mishra, Aleena Zahra, Lokendra V. Chauhan, Riddhi Thakkar, James Ng, Shreyas Joshi, Eric D. Spitzer, Luis A. Marcos, W. Ian Lipkin, Nischay Mishra

**Affiliations:** 1Center for Infection and Immunity, Mailman School of Public Health, Columbia University, New York, NY 10032, USA; mm5880@cumc.columbia.edu (M.M.); lc2910@cumc.columbia.edu (L.V.C.); rt2640@cumc.columbia.edu (R.T.); nj2208@cumc.columbia.edu (J.N.); sj3038@cumc.columbia.edu (S.J.); 2Division of Infectious Diseases, Department of Medicine, Stony Brook University, Stony Brook, NY 11794, USA; aleena.zahra@stonybrookmedicine.edu (A.Z.); eric.spitzer@stonybrookmedicine.edu (E.D.S.); luis.marcosraymundo@stonybrookmedicine.edu (L.A.M.)

**Keywords:** COVID-19, immunocompromised, SARS-CoV-2, infection, immune response

## Abstract

Immunocompromised individuals are at risk of prolonged SARS-CoV-2 infection due to weaker immunity, co-morbidities, and lowered vaccine effectiveness, which may evolve highly mutated variants of SARS-CoV-2. Nonetheless, limited data are available on the immune responses elicited by SARS-CoV-2 infection, reinfections, and vaccinations with emerging variants in immunocompromised patients. We analyzed clinical samples that were opportunistically collected from eight immunocompromised individuals for mutations in SARS-CoV-2 genomes, neutralizing antibody (NAb) titers against different SARS-CoV-2 variants, and the identification of immunoreactive epitopes using a high-throughput coronavirus peptide array. The viral genome analysis revealed two SARS-CoV-2 variants (20A from a deceased patient and an Alpha variant from a recovered patient) with an eight amino-acid (aa) deletion within the N-terminal domain (NTD) of the surface glycoprotein. A higher NAb titer was present against the prototypic USA/WA1/2020 strain in vaccinated immunocompromised patients. NAb titer was absent against the Omicron variant and the cultured virus of the 20A variant with eight aa deletions in non-vaccinated patients. Our data suggest that fatal SARS-CoV-2 infections may occur in immunocompromised individuals even with high titers of NAb post-vaccination. Moreover, persistent SARS-CoV-2 infection may lead to the emergence of newer variants with additional mutations favoring the survival and fitness of the pathogen that include deletions in NAb binding sites in the SARS-CoV-2 surface glycoprotein.

## 1. Introduction

Worldwide retrospective cohort studies report higher rates of hospitalization for critical care, severe disease, and increased mortality in immunocompromised COVID-19 patients [1]. A study with the Pfizer-BioNTech and Moderna vaccines in immunocompromised adults found anti-spike antibodies in 17% of participants after the first vaccine dose and 54% after the second dose [2,3]. In contrast, studies in immunocompetent adults report anti-spike antibodies in 90% of participants after the first vaccine dose and 100% after the second dose [4,5]. The risks of SARS-CoV-2 re-infections and breakthrough infections are higher in immunocompromised individuals compared to healthy individuals [6]. Published studies based on diverse cohorts of vaccinated populations suggested that booster doses of mRNA-based vaccines induce neutralizing antibodies towards the emerging SARS-CoV-2 Omicron variant [7]. Accordingly, public health agencies recommended a third dose of mRNA-based vaccines to boost immunity in all groups, and recently the Centers for Disease Control and Prevention (CDC) also recommended a fourth additional dose of an mRNA COVID vaccine in individuals who are moderately to severely immunocompromised to remain protective of infection [8]. Besides a hampered immune response, immunocompromised patients also showed persistent SARS-CoV-2 infections and can shed virus for longer periods compared to healthy individuals [9]. Several recent studies also suggested that such persistent infection might harbor several mutations and deletions in SARS-CoV-2 genomes that evolve to new variants in immunocompromised patients [10,11]. Published data also suggest that the recently emerged heavily mutated Omicron variant may have origipnated from a prolonged SARS-CoV-2 infection in an immunosuppressed individual with HIV [Media report 1: https://www.deseret.com/coronavirus/2021/12/27/22848061/omicron-variant-origins-where-omicron-came-from-started (accessed on 1 February 2022), Media report 2: https://www.npr.org/2021/11/30/1060185915/the-omicron-variant-might-have-originated-in-someone-with-a-suppressed-immune-sy (accessed on 1 February 2022)]. Thus, SARS-CoV-2 genome sequencing and variant analysis from immunocompromised patients is critical to monitor the emerging variants that lead to surges in COVID-19 cases due to immune escape after vaccination and previous infection.

In this study, we examined nasal swabs and sera samples collected retrospectively at different time points from a small group of eight COVID-19 patients with different immunosuppressive clinical conditions and performed rigorous molecular and serological analysis. We recovered full SARS-CoV-2 genome sequences from three patients and compared them for mutations in surface glycoproteins to the emerging variants. We also identified important deletions in the N-terminal domain (NTD) of surface glycoprotein. Deletions in the NTD are known to alter neutralizing antibody (NAb) binding sites. To the best of our knowledge, we are reporting for the first time the identification of immunoreactive epitopes for SARS-CoV-2 in immunocompromised COVID-19 patients using a high-density peptide array. Moreover, we determined NAb titers in these eight immunocompromised patients and discuss their immune responses in relation to treatment, reinfection, and vaccination.

## 2. Materials and Methods

### 2.1. Samples, Viral RNA Extraction, and Real-Time PCR

For this study, 21 clinical specimens, including 4 nasal swabs and 17 sera samples from eight immunocompromised patients with COVID-19 infection (Pts 1, 2, 3, 4, 5, 6, 7, and 8) were collected at multiple time points with informed consent (Stony Brook University, IRB code # 2021-00308) (Table 1). All patients were only positive with SARS-CoV-2 RNA at the time of admission or within 72 h of hospitalization using BioFire^®^ Respiratory 2.1 (RP2.1) Panel (Biofiredx). Viral RNA was extracted from nasal swabs in VTM or serum samples, and the SARS-CoV-2 viral load was determined from the Ct values using an FDA EUAapproved Triplex CII-SARS-CoV-2 rRT-PCR assay (Appendix A) [12].

### 2.2. SARS-CoV-2 Genomic Sequencing

Nearly complete SARS-CoV-2 genomic sequences (>99%) were recovered from the nasal swab samples of Pts 1, 6, and 7; the serum sample of Pt 1; and the cultured virus isolate from Pt 1 using a capture-based high-throughput sequencing (HTS) method (Appendix A). Illumina libraries were prepared using a KAPA-HyperPlus library preparation kit [13] and were enriched for SARS-CoV-2-specific viral inputs according to the Mybaits capture probes protocol (Daicel Arbor Biosciences, Ann Arbor, MI, USA). Purified and quantified libraries were then sequenced on a NextSeq 2000 Illumina platform. The reads were mapped against the SARS-CoV-2 Wuhan reference sequence (accession no. NC_045512) and a variant analysis was performed using Geneious R10 (https://www.geneious.com; accessed on 15 March 2022), GISAID [14], and NextClade [15]. The SARS-CoV-2 viral genome sequences were deposited to the GISAID COVID-19 data repository (Appendix A).

### 2.3. ELISA and End-Point Absolute Viral Neutralization Assay

Anti-spike-IgG and anti-NCP-IgG antibodies in sera samples were detected with commercial ELISA kits (EUROIMMUN) following the manufacturer’s instructions. Neutralizing antibody (NAb) titers in serum samples against the SARS-CoV-2 WA1, Alpha, Beta, Gamma, Delta, and Omicron variants and Pt 1 cultured virus (20A) were determined using an end point 100% viral neutralization assay [6]. The sera samples were diluted from 1:20 through 1:1280 dilutions for NAb titrations. Sera collected 1–4 weeks after the second dose of an mRNA-based vaccination (Pfizer-BioNTech) from two healthy adults were used as controls in the ELISA and end-point absolute viral neutralization assays (Table 2).

### 2.4. HCoV Peptide Array Experiment and Data Analysis

Seventeen serum samples from eight patients were analyzed using a high-density HCoV peptide array [16] as described previously [13,17,18,19,20]. The HCoV peptide array represents the proteomes of known HCoVs, including all variants of SARS-CoV-2, using 12 aa peptides that tile those proteomes with 11 amino acid overlap. For background correction and threshold generation, 1000 random scrambled nonspecific peptides were included. Six serum samples from individuals that had recovered after COVID-19 were used as positive controls. Fluorescent signal data for all the peptides from the IgG and IgM scanned images of all the HCoV peptide arrays were converted to arbitrary units (AU), pooled, background corrected, and normalized to avoid inter-experimental variation [17,18,20]. A peptide signal was considered reactive if the intensity reading (AU) was above the threshold (mean ± 2 SD readings of random peptides, >7500 AU for IgG and >5000 AU for IgM analysis). A cut-off threshold for peptide recognition was defined as mean ± 2 times the standard deviation (SD) of the mean intensity value of all random scrambled nonspecific peptides [21]. An epitope sequence was reassembled from these filtered immunoreactive peptides. Binary calls for the presence or absence of immunoreactivity in these epitopes were performed. A sample was considered positive for an epitope if three consecutive peptides within that region had intensities above the defined threshold.

## 3. Results

### 3.1. Clinical Summary of Patients with SARS-CoV-2 Infections and Re-Infections

Details of the clinical history of the eight immunocompromised COVID-19 patients with severe disease are reported in Table 1, where four had fatal outcomes (Pts 1, 3, 5, and 7). Fatalities were predominantly due to COVID-19-triggered cardiac failure or multi-organ failure (Table 1). All the patients were hospitalized due to severe illness with COVID-19 and had lymphopenia. In the presence of lymphopenia, the results obtained from immune marker tests, such as total immunoglobulin levels or the CD4 count, are not reliable; therefore, these tests were not performed [22,23]. No patient had any other viral infection or concurrent bacteremia at the time of admission; however, Pt 3 and Pt 7 developed bacterial pneumonia during their prolonged hospitalization and were treated with antibiotics.

The erythrocyte sedimentation rate (ESR), C-reactive protein (CRP), white blood cell (WBC), and absolute lymphocyte counts were recorded at the time of admission and the last day of hospitalization for all the patients (Appendix A). The CRP levels at the time of hospitalization were higher in deceased immunocompromised patients compared to the COVID-19-recovered individuals. Higher procalcitonin (PCT) levels were detected in patients with repeated SARS-CoV-2 infection, including Pt 1 (~40-fold above the cutoff) and Pt 3 (~350-fold above the cutoff) at the time of death (Appendix A). Computerized tomographic thoracic imaging of fatal cases (Pts 1, 3, 5, and 7) showed increased multifocal consolidative opacities in the lungs, which is consistent with COVID-19 pneumonia (Appendix A). CT scans of recovered cases (Pts 2, 4, 6, and 8) revealed emphysematous changes and scattered areas of ground glass opacities.

**Table 1 viruses-14-00934-t001:** Demographic and clinical conditions of immunocompromised COVID-19 patients under study.

Patient IDs	Sex (~Age Range in Years)	Latest Hospitalization Date *	COVID-19 Symptoms	COVID-19 Outcome (Days Since Hospitalization)	Immunocompromised Conditions	Immunosuppressed Treatment	Other Co-Morbidities
Pt 1	M (70–74)	2 April 2021	Hemoptysis, persistent high fevers, diarrhea, cough	Died (17 d) due to cardiac arrest as a complication of COVID-19 infection	Diffuse large B-cell lymphoma (DLBCL)	Rituximab	Hypertension, non-ST-elevation myocardial infarction, coronary artery disease (CAD), coronary artery bypass graft (CABG)
Pt 2	F (50–54)	14 April 2021	Fever, nausea, chills, severe headache	Recovered (5 d)	Renal transplant (02/2021) and Neutropenia	Tacrolimus, Mycophenolate, plasmapheresis, IVIG	Hypothyroidism, DM2, hypertension
Pt 3	F (60–64)	4 April 2021	Shortness of breath, malaise	Died (14 d) due to multiorgan failure (cardiopulmonary, hepatic, renal), acute cerebrovascular accident (CVA)	Chronic obstructive pulmonary disease (COPD)	NA	Opioid use, alcohol use, hypothyroidism
Pt 4	F (30–34)	14 April 2021	Fatigue	Recovered (4 d)	Systemic lupus erythematous (SLE) and lupus nephritis	High-dose Prednisone	Sickle cell trait, hypertension
Pt 5	M (55–59)	29 March 2021	Shortness of breath, dyspnea, cough, diarrhea, fever, chills	Died (10 d) due to multiorgan failure (renal, pulmonary, cardiac), acute CVA	Leukemia and bone marrow transplant (twice in 2014 and 2016)	Ruxolitinib, Methotrexate	Hypertension, asthma
Pt 6	F (60–64)	29 March 2021	shortness of breath, cough, fatigue	Recovered (10 d)	Follicular lymphoma	R-CHOP (monoclonal antibody rituximab with cyclophosphamide, doxorubicin, vincristine, and prednisone)	Hypertension, asthma
Pt 7	F (65–69)	30 March 2021	fever, myalgia, loss of taste, and tachypnea	Died (14 d) due to multiorgan failure (cardiac, pulmonary)	Pulmonary hypertension	NA	Hypothyroidism, hypertension, history rheumatic fever
Pt 8	F (80–84)	28 March 2021	Malaise, sore throat, and cough	Recovered (11 d)	Breast cancer	Exemestane (aromatase inhibitors)	Nonischemic cardiomyopathy, s/p implantable cardioverter-defibrillator (ICD), hypertension

* SARS-CoV-2 PCR-positive date for the latest infection. (NA—Not analyzed).

A timeline of each patient’s clinical course and hospitalization is shown in Figure 1. Pt 1 was hospitalized in April 2021 but visited hospital multiple times during January–March 2021 with COVID-19-like symptoms after recovery from a first SARS-CoV-2 infection in November 2020. Out of all immunocompromised patients, only Pt 1 received convalescent plasma therapy during treatment (Figure 1). Other immunocompromised patients (Pts 2, 3, and 4) had first SARS-CoV-2 infections in early 2020 and recovered without hospitalization but were hospitalized in April 2021 after reinfection. Pt 2 was treated with Remdesivir for 5 days and received empiric ceftriaxone for 7 days before recovering (Figure 1). A second fatal case, Pt 3, received steroids but not antivirals or monoclonal antibodies. Pt 4 was discharged after 4 days of hospitalization without receiving specific COVID-19 management or respiratory support.

Patients 5, 6, and 7 were hospitalized with severe COVID-19 symptoms a week after the second dose of an mRNA vaccine. Patient 8 received only one dose of an mRNA vaccine. Patients 5 and 7 both required mechanical ventilation and died after 10 and 14 days of hospitalization, respectively (Table 1). Patient 6 recovered 10 days post-hospitalization without any mechanical ventilation. Treatment with immunosuppression medication (Exemestane) was temporarily stopped for Pt 8 for 10 days. Patient 8 was discharged on supplemental oxygen after hospitalization for 22 days (Figure 1).

### 3.2. SARS-CoV-2 Genome Analyses

In total, 7 of 21 available samples (4 nasal and 17 sera samples) from eight patients tested positive for SARS-CoV-2 RNA using a CII-Triplex-SARS-CoV-2 rRT PCR assay (Appendix A) [12]. SARS-CoV-2 RNA was present in the nasal swabs of Pt 1 (Ct 22), Pt 5 (Ct 43), Pt 6 (Ct 35), and Pt 7 (Ct 32) and serum samples of Pt 1 (Ct 31) and Pt 5 (Ct 36). Near complete SARS-CoV-2 genomic sequences (>99% genome recovery) recovered from Pt 1, Pt 6, and Pt 7 were submitted to the GISAID database (accession ID: EPI_ISL_4298277 to EPI_ISL_4298281). The SARS-CoV-2 genome sequences of different variants of concerns, additional Alpha and Delta variant sequences recovered from immunocompetent individuals from recently published studies [24,25], and the viral genomes recovered from the immunocompromised patients in this study were compared for their genomic variability (Figure 2A). The genome recovered from Pt 1 represented the 20A variant with characteristic F4V, L212V, L452R, D614G, A672V, and P681H spike aa substitutions. The viral genome from Pt 6 was of the Alpha (20I, B.1.1.7) variant with characteristic N501Y, A570D, D614G, P681H, T716I, S982A, and D1118H spike aa substitutions. The genome from Pt 7 was of the Iota (21F, B.1.526) variant with characteristic L5F, T95I, D253G, S477N, D614G, and A701V aa mutations in the surface glycoprotein (Figure 2A). A 24-nucleotide (8 aa) deletion within the N-terminal domain (NTD) of the surface glycoprotein (nt position: 21,973–21,997, aa position: 138–145) was detected in Pt 1 (20A variant) and Pt 6 (Alpha variant) (Figure 2B,C) and confirmed by PCR and Sanger sequencing. The L212V mutation was detected in Pt 1 (20A variant) at an Omicron-specific characteristic mutation site (L212I) in the NTD of the surface glycoprotein (Figure 2A). Pt 6 also had the Alpha-variant-specific aa deletion (del 69,70 aa) in the recurring deletion region 1 (RDR1) of the NTD in the surface glycoprotein.

To test the viability and infectivity of the 20A and Alpha variant viruses with 8 aa deletions, we pursued a virus culture in Vero E6 cells. Only the virus from Pt 1 was successfully propagated, presumably due to a higher viral load (Ct 22). We recovered complete genomic sequences from the Pt 1 virus culture (Pt1_VC) and confirmed the 24-nucleotide (8 aa) deletions in the NTD of the surface glycoprotein (Figure 2B,C). The growth curve kinetics of the cultured virus (20A) was similar to five other SARS-CoV-2 variants (WA1, Alpha, Beta, Gamma, and Omicron) (Figure 2D).

### 3.3. Serological Analyses

#### 3.3.1. ELISA Assays

Serum samples collected at different time points during hospitalization were screened for anti-spikeIgG and anti-NCP-IgG antibodies using commercial ELISA assays (EUROIMMUN). Serum samples of Pts 5, 7, and 8, who received mRNA-based vaccines just one week before hospitalization, had anti-spike-IgG and anti-NCP-IgG antibodies. Serum samples from other immunocompromised patients (Pts 1, 2, 3, 4, and 6) lacked anti-spike-IgG antibodies. Only one out of eight immunocompromised patients, Pt 1, had anti-NCP-IgG antibodies (Table 2).

#### 3.3.2. End-Point Absolute Viral Neutralization Assay

Neutralizing antibodies (NAb) against the prototypic Washington strain (WA1), five variants of concern (Alpha, Gamma, Beta, Delta, and Omicron), and the cultured 20A virus from Pt 1 were measured using an end-point 100% viral neutralization assay [6]. NAb titers against WA1 variants were only present in three vaccinated immunocompromised patients (Pts. 5, 7, and 8) but were absent in the others (Pts 1, 2, 3, 4, and 6). The highest NAb titers were detected in Pt 5 (deceased) and Pt 8 (recovered) (Table 2). NAb titers in vaccinated immunocompromised COVID-19 patients were highest against the SARS-CoV-2 WA1 variants followed by the Alpha, Gamma/Beta, and Delta variants (Table 2). No NAb titers were detected in any patients against the Omicron variants. Vaccinated immunocompromised patients had reduced NAb titers, whereas non-vaccinated immunocompromised patients had no NAb titers against the 20A cultured virus from Pt 1 (Table 2). Higher NAb titers were measured against prototype WA1 and the cultured 20A variant in vaccinated individuals (CON-0132 and CON-0133) with healthy immune systems, and lower NAb titers were measured against the other variants.

**Table 2 viruses-14-00934-t002:** Anti-spike and anti-nucleocapsid EUROIMMUN ELISA assay and neutralizing antibody titers tested against different SARS-CoV-2 variants.

			EUROIMMUN ELISAs		Neutralizing Antibody Titers against Different SARS-CoV-2 Variants						
Patient ID	COVID-19 Outcome	Serum Collected (Days Post-Infection)	Surface Glycoprotein	Nucleocapsid	WA1 Variants	Alpha (B.1.1.7)	Gamma (P.1)	Beta (B.1.351)	Delta (B.1.617)	Omicron (B.1.1.529)	20 A (Pt 1)
Pt 1	Fatal	04	Negative	Positive	≤1:20	≤1:20	≤1:20	≤1:20	≤1:20	≤1:20	≤1:20
		10	Negative	Positive	1:80	≤1:20	≤1:20	≤1:20	≤1:20	≤1:20	≤1:20
		17	Negative	Positive	≤1:20	≤1:20	≤1:20	≤1:20	≤1:20	≤1:20	≤1:20
Pt 2	Survival	0	Negative	Negative	≤1:20	≤1:20	≤1:20	≤1:20	≤1:20	≤1:20	≤1:20
		05	Negative	Negative	≤1:20	≤1:20	≤1:20	≤1:20	≤1:20	≤1:20	≤1:20
Pt 3	Fatal	10	Negative	Negative	≤1:20	≤1:20	≤1:20	≤1:20	≤1:20	≤1:20	≤1:20
		14	Negative	Negative	≤1:20	≤1:20	≤1:20	≤1:20	≤1:20	≤1:20	≤1:20
Pt 4	Survival	0	Negative	Negative	≤1:20	≤1:20	≤1:20	≤1:20	≤1:20	≤1:20	≤1:20
		04	Negative	Negative	≤1:20	≤1:20	≤1:20	≤1:20	≤1:20	≤1:20	≤1:20
Pt 5	Fatal	05	Positive	Positive	1:320	≤1:20	≤1:20	≤1:20	≤1:20	≤1:20	≤1:20
		10	Positive	Positive	1:1280	1:640	≤1:20	1:40	≤1:20	≤1:20	1:80
Pt 6	Survival	05	Negative	Negative	≤1:20	≤1:20	≤1:20	≤1:20	≤1:20	≤1:20	≤1:20
		10	Negative	Negative	≤1:20	≤1:20	≤1:20	≤1:20	≤1:20	≤1:20	≤1:20
Pt 7	Fatal	09	Positive	Positive	1:160	≤1:20	≤1:20	≤1:20	≤1:20	≤1:20	1:80
		14	Positive	Positive	1:640	≤1:20	1:40	≤1:20	≤1:20	≤1:20	1:160
Pt 8	Survival	17	Positive	Positive	1:1280	1:160	1:320	≤1:20	≤1:20	≤1:20	1:40
		22	Positive	Positive	1:640	1:320	1:640	1:80	≤1:20	≤1:20	≤1:20
CON-0132	Survived	37	Positive	Negative	1:40	≤1:20	NA	NA	1:40	<1:20	1:320
CON-0133	Survived	8	Positive	Negative	1:80	1:160	NA	NA	1:80	1:40	1:160

(NA—Not analyzed).

#### 3.3.3. Peptide Array Analysis

Antibodies to linear B-cell epitopes were identified, and immunoreactivity was evaluated using a high-density programmable HCoV peptide array [13,16,17,18,19,20]. A total of 264 immunoreactive IgG epitopes were identified. It was observed that 22 of 264 epitopes (11 in the surface glycoprotein, 7 in ORF1ab polyprotein, and 1 each in ORF3a, ORF8, and the membrane and nucleocapsid protein regions) were more immunoreactive with sera from immunocompromised patients and controls (Figure 3). IgG epitopes were more frequently reactive in recovered than in deceased reinfection patients. Epitope S-IgG-Pep 1 was immunoreactive in all positive controls and serum from all immunocompromised patients except the sera collected at the first time-point from Pt 8. Epitope S-IgG-Pep8 in the spike region (GAENSVAYSNNSIAIPTNFTI) was only immunoreactive in recovered patients (Figure 3) and was not immunoreactive in any of the positive controls. Epitope ORF1ab-IgG-Pep12 (QKLLKSIAATRGATVVIGTS) was reactive in all immunocompromised patients (100%) with reinfections and in only one vaccinated immunocompromised patient (Pt 6). Three epitopes in the surface glycoprotein (S-IgG-Pep9, S-IgG-Pep10, and S-IgG-Pep11) were immunoreactive in all positive controls and deceased vaccinated patients only. We and others have identified two epitopes in the surface glycoprotein with neutralizing-antibody-binding potential (S-IgG-Pep3 and S-IgG-Pep6) [16,26]. Epitope S-IgG-Pep3 was immunoreactive in Pts 1, 2, 4, 5, and 7. Immunoreactivity in Pt 1 was found only after convalescent plasma therapy; thus, we cannot exclude the possibility that this reflected antibodies in the donor plasma (Figure 3). Epitope S-IgG-Pep6 was not immunoreactive in any deceased reinfection patients (Pts 1 and 3) or recovered vaccinated patient (Pt 6).

In the IgM data analysis, 10 epitopes (9 from ORF1ab polyprotein and 1 from membrane glycoprotein) were immunoreactive with antibodies from three of four vaccinated immunocompromised patients (Pts 5, 7, and 8). In vaccinated immunocompromised patients with COVID-19 infections, increased IgM reactivity to spike epitopes were more frequent in deceased than in recovered patients (Figure 3). No IgM immunoreactivity was found in serum from re-infected patients (Figure 3). A membrane glycoprotein epitope M-IgG-Pep22 and M-IgM-Pep1 (MADSNGTITVEELKKLLEQWNLV) was immunoreactive with both IgG and IgM antibodies in all vaccinated immunocompromised patients and one patient with reinfection (Pt 1).

## 4. Discussion

Approximately 10 million Americans are immunocompromised and living with debilitating underlying co-morbidities and immunosuppressive medications. Almost all of these immunocompromised individuals have received two doses of an mRNA vaccine, and CDC recommended a third booster as well as a fourth protective dose in immunocompromised patients. Nonetheless, breakthrough infections and re-infections are continuously reported in this population [27,28]. In an effort to further understand if these infections could be linked to an inefficient antibody response and the implications of infection for the evolution of variant viruses, we analyzed opportunistically collected samples from eight immunocompromised patients with severe COVID-19 who were admitted to Stony Brook Hospital. We sequenced nasal samples for variant analysis and profiled humoral antibody responses in patient sera samples.

Patients were either hospitalized with lymphocytopenia or lymphocytopenia developed during hospitalization. Immuno-suppressive marker tests such as total immunoglobulin levels or CD4 counts were not performed because lymphocytopenia may result into suboptimal results, which could not be used in clinical management [22,23]. Pts 3, 4, 6, 7, and 8 were on steroid-based regimens that can interfere with normal immune function by different mechanisms, including the sequestration of CD4^+^ T Lymphocytes [29,30]. Pt 1 and Pt 6 were maintained on rituximab, an anti-CD20 inhibitor that depletes the B-cell-mediated immune response; Pt 2 was maintained on Tacrolimus and Mycophenolate mofetil, which inhibits the proliferation of T and B lymphocytes, leading to suppression of both humoral and cell-mediated immune responses [31]. Pt 5 was on Ruxolitinib (Jakafi) and Methotrexate. Ruxolitinib is known to suppress the innate and adaptive immune system and impair T-cell lymphocyte function [32,33]. Pt 8 was receiving the aromatase inhibitor Exemestane for estrogen-receptor-positive breast cancer. Estrogen modulates and upregulates immune responses via Toll-like receptors [34]. The temporary suspension of the Exemestane for 10 days may have enabled sufficient recovery of immune function in Pt 8 for viral clearance and recovery.

Seven out of eight immunocompromised patients were >50 years of age [35]. The levels of COVID-19 severity clinical markers (ESR, WBC count, CRP, and PCT) were elevated in all patients [36,37,38]. Patients 1 and 5, who died, also had SARS-CoV-2 RNA in the blood, suggesting a compromised antiviral immune response [39] or a failure to develop immunity after mRNA vaccination [40] preventing clearance of the virus. SARS-CoV-2 RNA was present 90 days after the first infection in Pt 1, and previous studies suggested that a poor immune response can prolong virus shedding [9,41]. Pt 1 was admitted three times to intensive care units with COVID-19 symptoms within a 5-month period, and Pts 2, 3, and 4 were first infected with SARS-CoV-2 in mid-2020 and re-infected in 2021. These reinfections led to symptomatic severe COVID-19 disease, suggesting that immunocompromised individuals develop weaker immunity after infection that can increase the risk of severe disease with re-infections in comparison to individuals with healthy immune systems [42].

We recovered SARS-CoV-2 viral genomes representing three different variants (20A, 20I (Alpha), and 21F (Iota)). The presence of unique additional mutations (F4V, I468V, N354D, F490S, and A672V) in the 20A viral genome recovered from Pt 1 (Figure 2A) suggests that a lowered immunity in such immunocompromised individuals creates a favorable environment for the virus to mutate [9]. The presence of non-synonymous mutations (N501Y and A570D) in the receptor binding domain (RBD) of the surface glycoprotein from the SARS-CoV-2 genome of Pt 6 and the well-established escape mutations P681H, T716I, T95I, D253G, S477N, and A701V from the SARS-CoV-2 genome of Pt 7 confirm the potential for the emergence of viral mutations in immunocompromised patients, irrespective of strain/variant, that enable breakthrough and reinfections due to the evasion of adaptive immune responses [43].

Three of four patients with SARS-CoV-2 reinfections did not have anti-S-IgG or anti-NCP-IgG antibodies. A serum sample collected from Pt 1 three days after receiving convalescent plasma therapy had low titer anti-S-IgG and anti-NCP-IgG antibodies in ELISA and NAb against the WA1 variant of SARS-CoV-2. Although anti-NCP antibodies persisted, anti-spike and NAbs were not present in the plasma one week later. Pt 1 had moderate titers of NAb but died after reinfection. Pt 6, with characteristic RBD-specific spike mutations and NTD deletions, lacked anti-spike, anti-NCP antibodies and had low NAb titers, even after full vaccination, but survived, suggesting NAb titers are not correlated with survival, and other protective mechanisms such as T-cell-mediated immunity [44] may also play significant roles along with the humoral response. Four of the eight patients were discharged after full recovery from COVID-19 in March-April 2021. A follow-up in July-August 2021 found that none of the recovered patients had any respiratory complications or signs of long COVID (post-acute sequelae of SARS CoV-2 infection, PASC).

We found high NAb titers against the WA1 variant in vaccinated immunocompromised patients. This likely represents the use of prototype SARS-CoV-2 sequences for vaccine design [45,46]. The exception was Pt 6, who had low NAb titers but was infected with the Alpha variant and had an 8 aa deletion within the NTD of the surface glycoprotein. Virus cultured from Pt 1 (20A) was genetically similar to the prototype WA1 variant, except for the 8 aa deletions in the NTD of the surface glycoprotein. The latest Omicron SARS-CoV-2 variant also had this crucial three aa deletions from 143–145 in the NTD of the surface glycoprotein, which highlighted the impact of genetic changes on the observed virulence and immune evasion property of this virus. The cultured virus isolate was infectious and had growth kinetics similar to the WA1 variant; however, NAbs against this virus in three immunocompromised vaccinated patients (Pts 5, 7, and 8) were reduced by many fold, probably due to importance of the NTD in viral neutralization [47]. On the contrary, vaccinated immunocompetent individuals had higher NAbs against cultured virus 20A with the 8 aa deletion in the NTD (from Pt 1), suggesting multiple NAb binding sites in immunocompetent individuals and also that immunocompromised patients fail to develop an efficient immune response after two doses of mRNA vaccination (https://www.healthline.com/health/vaccinations/which-covid-vaccine-is-best, accessed on 31 March 2022). We found no NAbs against the 20A isolate and Omicron variant in immunocompromised patients with reinfections.

The peptide array data showed immunogenic epitopes through the SARS-CoV-2 proteome in all 17 sera samples collected from eight patients, including sera samples, which tested negative in ELISA and neutralization assays. These immunoreactive epitopes can be used in serological assays to screen past SARS-CoV-2 exposure and vaccinations in immunocompromised individuals. We observed highly immunoreactive epitopes to both structural (spike, nucleocapsid, and membrane glycoprotein) and non-structural (Orf1ab, Orf3a, and Orf8) viral proteins. Of all immunoreactive epitopes detected, one within the membrane glycoprotein had highest sensitivity in both the IgG and IgM responses [48]. Fatal patients had lower immunoreactivity to epitopes from surface glycoprotein and ORF1ab polyproteins than recovered patients (Figure 3). Irrespective of measured NAb titers against different VOCs and the status of vaccination, four out of eight immunocompromised patients survived the COVID-19 infections without any lingering symptom after recovery. Survival could be dependent on broader protective immunological mechanisms such as T-cell mediated immunity in such patients, apart from immunoglobulin-mediated responses [44].

We acknowledge that our data are based on a small opportunistic observational study; nevertheless, our observations add to our understanding of the pathogenesis and prognosis of COVID-19 in immunocompromised individuals. In summary, we observed that (1) the risk of a fatal outcome in immunocompromised individuals with SARS-CoV-2 infections may be increased with the use of immunosuppressive drugs, (2) infections may occur in individuals despite high titers of NAb, (3) persistent infection may lead to the emergence of viral variants with deletions in an NAb binding site in the SARS-CoV-2 surface glycoprotein that result in escape from vaccination-induced immunity, and (4) new variants with excessive mutations may arise from persistent SARS-CoV-2 infection in immunocompromised individuals.

### Limitations of the Study

A significant limitation to this study of immunocompromised COVID-19 patients was the lower sample size, limited number of patients, and inadequate sampling. We were constrained with the recovery of a complete viral genome from all the immunocompromised patients since the viral load was too low to use the sample for the high-throughput sequencing. Indeed, a thorough sample collection at multiple time points of infections of these immunocompromised patients would have enabled us to perform longitudinal testing for virus evolution and immune responses. Besides studying humoral responses, an evaluation of T-cell responses in these immunosuppressed patients could be critical to better understand the immune responses to vaccination and previous infections, but we were constrained with a retrospective study design. In future studies, the effect of 3rd booster doses and additional 4th protective mRNA-based vaccine doses in immunocompromised patients will also provide additional data.

## Figures and Tables

**Figure 1 viruses-14-00934-f001:**
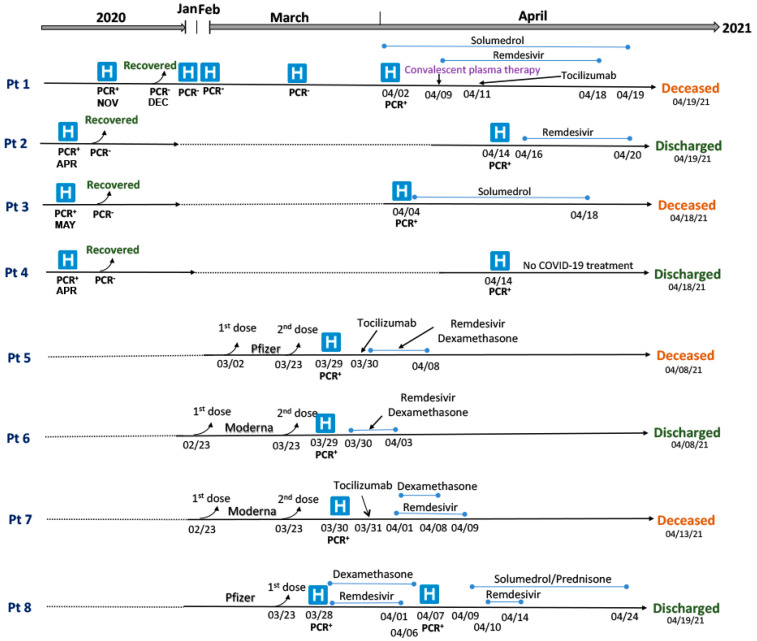
Timeline of immunizations, SARS-CoV-2 infection, hospitalization (H), and treatment of eight COVID-19 immunocompromised patients.

**Figure 2 viruses-14-00934-f002:**
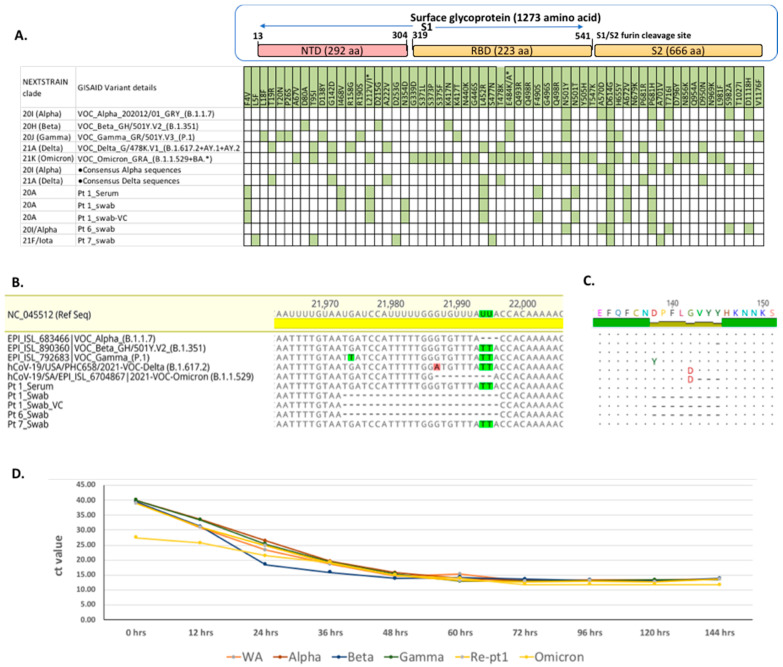
Complete genome analysis of SARS-CoV-2 genomes recovered from Pt 1, Pt 6, and Pt 7. (**A**) Nucleotide alignment of surface glycoprotein gene recovered from Pt 1, Pt 6, and Pt 7 showing non-synonymous nucleotide mutations compared against other SARS-CoV-2 variants of concern (VOCs) and the consensus Alpha and Delta sequences from previous studies. •Consensus Alpha variant sequence was generated from alignment of 88 individual SARS-CoV-2 Alpha variant genomic sequences recovered from 88 patients [24], and consensus Delta variant sequence was generated from alignment of 44 individual SARS-CoV-2 Delta variant genomic sequences recovered from 44 patients [25] using Geneious R10 software. (**B**) Twenty-four nucleotide deletions were observed in the surface glycoprotein of the virus recovered from Pt 1-Swab, Pt 1 Swab-VC, and Pt 6 swab; (**C**) Eight amino acid deletions from 138–145 positions were observed in the protein alignment; (**D**) Time-point growth curve for SARS-CoV-2 VOCs (WA1, Alpha, Beta, Gamma, and Omicron) and Pt 1_VC from 0 to 144 h in vitro.

**Figure 3 viruses-14-00934-f003:**
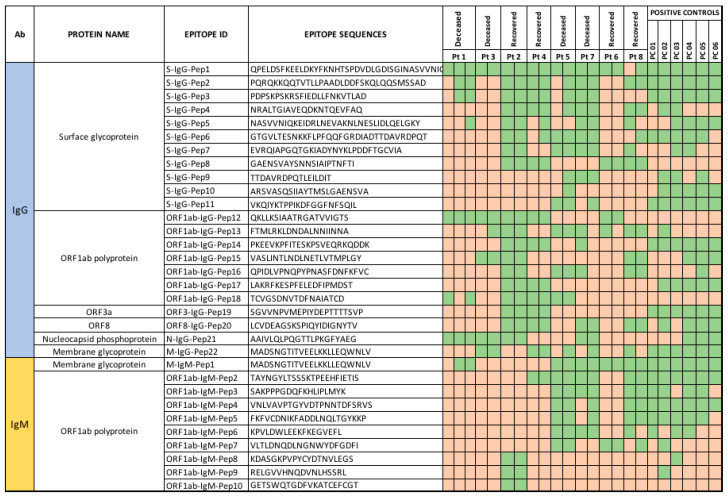
Checkerboard representing both IgG and IgM immunoreactivity of SARS-CoV-2 epitopes across the polyproteins in eight immunocompromised patients and six immunocompetent COVID-19-recovered individuals (positive controls). (In the heat map, green cell color: immunoreactivity, orange cell color: no immunoreactivity to the epitopes for each sample).

## Data Availability

The data analyzed in this study can be made available upon request to the corresponding author.

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
