# Peer review of "A Short Series of Case Reports of COVID-19 in Immunocompromised Patients"

_viruses, 2022, doi:10.3390/v14050934_

Round 1

Reviewer 1 Report

This reviewer congratulates the authors for this "opportunistic observational study". They did a fair job to present important facets of COVID-19 in immunocompromised patients. They showed that neutralising antibody levels are generally low in these patients, and also showed that there is no strong correlation between antibody dependent immunity and survival. Having saind that this reviewer has some suggestions to increase the potential impact of the work.

  1. It would be good to discuss the fact, that these patients seems to be infected repetitively by the SRAS-CoV-2 and the outcome of the infections seems to not be affected by previous recovery from the disease.
  2. Fig. 2A shows a great variety of genetical variability. It would be nice to compare this variability to patients without immunocompromising medical treatments. These patients may come from previously published relevant articles or from their own patients (if they have access to such patients materials). Direct comparison between these groups would improve the novelty and impact significantly.
  3. The discussion should be amended with the fact, that neutralising antibodies did not show correlation with survival. Patient 5 had fatal outcome with high neutalizing antibody titers, while Patient 8 survived. Additionally, no neutralizing antibodies were detected in many cases, and some patients survived, others died. The discussion should address the potential immonological mechanisms besides to immunoglobulin mediated responses, which can explain these findings. In addition, the authors are encouraged to speculate the effectiveness of T-cell mediated immunity in these patients.
  4. Authors need to discuss that fatal outcome may be the result of infections (independent of SARS-CoV-2), which can shift the balance of COVID-19 to a less desirable outcome. Some of the patients may died as a result of additional infections, and COVID-19 "only" contributed to have a weakened general health condition.
  5. Finally, this reviewer suggests to discuss if immunization by mRNA based vaccines can get the desired effect in these patients. It needs to be based on the observed occurrence of neutralizing antibodies: it appears that vaccination did not yield a reliable neutralizing antibody production. This can be less significant than that in individuals without immuncompimising drug treatments. Discussion may address the issue if vaccination is beneficial and how in immunocompromised patients.

Reviewer 2 Report

The paper by Mishra and co-workers  deals with a very relevant aspect of SARS CoV-2 infection: the response in immunocompromoised patients.

Unfortunately this study has many severe drawbacks that hamper the scientific soundness of the manuscript.  Please find  following a list of the weakness points that must be addressed by the Authors.

  1. As correctly underlined by the Authors themsleves the styudy included  only 8 patients and this number is by far too low to allow wany kind of conclusion.
  2. In addition the 8 patinets suffered from really different clinical conditions that  caused their immunosuppression and no onformation are provided about the measurable level of this immune suppressive status.
  3. As a consequence of the very different basic pathological condition that affected each single patients  many different therapeutic regimens wera in place with obvious diversity in the mechanisms of immune suppression
  4. No information is provided about the final, cause of each patient death: did they died FOR  SARS CoV-2 infection or WITH  COVID-19?
  5. The overall methodologiues applied to study the immune response to SARS CoV-2 are of valòue but in the case of immunosuppressed patients an evaluation of the T-cell response (even by using a IGRA test) is of paramount relevance in order to have a complete picture of the immune reaction against SARS CoV-2  and (eventually) vaccinations.
  6. The sequencing (WGS) data were available only for a subset of the 8 patients and this is of course making the data  of very low interest. Which NGS system was used for sequencing?
  7. It would hvae been of great interest to know which NAb titer  the sera showed against the SARS CoV-2 isolated from each homologous patient, given the very low variability detected for most of the patients against  the VOCs studied.

Round 2

Reviewer 2 Report

I thank the Authors for the revised version of their paper. I understand the many difficulties that they encountered given the limted numebr of retrospectively available samples. These are now well underlined in the text. Being  clearly defined by the Authors that this is just an obesrvational retrospective  study, I strongly suggest that the title could be modified accordingly in order to make the reader aware from the first look about the nature of the study. One possible suggestion could be: "A short series case report of COVID 19 in immunocompromised patients". Thsi woul dmek immediately clear that no overall conclusion can be taken and that this is just  a description of a short series of differently immunosupressed subject affected by SARS CoV-2 infection.

Author Response

We are very very thankful to the reviewer for responding to our revised manuscript.

We agree that a new title will provide additional clarity to readers so we have modified the article title as reviewer suggested to the following:

"A short series case reports of COVID 19 in immunocompromised patients"